# Spatial distribution and determinants of physical intimate partner violence among women in Kenya: Evidence from the 2022 Kenya Demographic and Health Survey

Joshua Okyere[1,2,3], Castro Ayebeng [1,3]*, Ebenezer N. K. Boateng[4], Rebecca A. A. Assie[5], Amanda Odoi[6], King-David Dzirassah[1], Bright Ankomahene[7], Kwamena Sekyi Dickson[1]

1 Department of Population and Health, University of Cape Coast, Cape Coast, Ghana, 2 Department of Nursing, College of Health Sciences, Kwame Nkrumah University of Science and Technology, Kumasi, Ghana, 3 Department of Research and Advocacy, Challenging Heights, Winneba, Ghana, 4 Department of Geography and Regional Planning, University of Cape Coast, Cape Coast, Ghana, 5 Department of Education and Psychology, University of Cape Coast, Cape Coast, Ghana, 6 Centre for Gender Research, Advocacy and Documentation (CEGRAD), University of Cape Coast, Cape Coast, Ghana, 7 Department of Geomatics Engineering, College of Engineering, Kwame Nkrumah University of Science and Technology, Kumasi, Ghana

* castro.ayebeng@stu.ucc.edu.gh

**Data Availability Statement:** This study used a publicly existing database of the Demographic and Health Survey which is freely accessible for

## Abstract

### Background

Despite global, regional, and national efforts to address intimate partner violence (IPV), physical IPV persists as a significant challenge in Kenya. This study employs geospatial analysis to examine the spatial distribution and determinants of physical intimate partner violence among women, aiming to inform targeted interventions and policies.

### Methods

The study used a secondary analysis of a cross-sectional study design based on the 2022 Kenya demographic and health survey. Analyses were conducted using Stata version 17.0 and ArcMap version 10.8. Spatial autocorrelation and hotspot assessment were conducted in the geospatial analysis, while a multilevel logistic regression model was used to examine determinants of physical violence among reproductive-aged women.

### Results

The study found 28.8% (10,477) of the surveyed women reported experiencing physical intimate partner violence. The spatial analysis identified significant clusters in the southwest and central regions, with women in sub-counties like Chepaluugu, Konion, Sotik, Bumula, and Metayos among others experiencing more violence. Conversely, women in areas in the North East and South East corners such as Kisauni, Tarabaj, Waijir North, Lafey, and Mandera North and South among others showed little or no physical intimate partner violence. Multivariable logistic regression identified age, education, wealth index, partner domineering indicators, and justification of wife beating to be associated with physical intimate partner

download using the following link "https://dhsprogram.com/data/dataset/Kenya_Standard-DHS_2022.cfm?flag=0.

**Funding:** The author(s) received no specific funding for this work.

**Competing interests:** The authors have declared that no competing interests exist.

**Abbreviations:** GBV, Gender-based violence; IPV, Intimate partner violence; VAW, Violence Against Women; KDHS, Kenyan Demographic and Health Survey; GIS, Geographic Information Systems; AOR, Adjusted Odds Ratio.

violence. Higher education and wealth were associated with lower violence odds, while partner domineering indicators and justification of wife beating increased odds.

## Conclusion

Spatial variations in intimate partner violence risk for women in Kenya underscore the need for targeted government interventions. Focusing on hotspot regions, especially among women with the poorest wealth index, no formal education, and older age, is crucial. Implementing behavior change campaigns addressing violence justification and partner dominance is vital. Active involvement of male partners in programs aiming to eliminate intimate partner violence is essential for comprehensive impact.

## Introduction

Intimate Partner Violence (IPV), which constitutes a phenomenon where an individual is subjected to physical, sexual and emotional abuse by a person they are currently or have ever been in a relationship with, has, over the years, continued to remain a major global and public health challenge [1–3]. Physical intimate partner violence constitutes a significant proportion of the types of intimate partner-partner violence. It involves the subjection of an individual's sexual or intimate partner to physical assaults such as slapping, kicking, strangulation and hitting [4]. Globally, approximately 30 percent of women have been exposed to intimate physical violence [5]. Evidence from 26 sub-Saharan African countries identifies physical violence (30.58%) as the major IPV suffered by women followed by emotional (30.22%) and sexual IPV (12.26) [5]. In the context of Kenya, approximately 35 percent of women experience physical violence orchestrated by their intimate partners [6].

At both the international and national front, it is a consensus rallying around the elimination of all forms of violence including physical IPV by the end of 2030 [7]. This call is premised on evidence of a strong association between physical IPV and several adverse health effects and social risks. For instance, physical IPV has been reported to significantly increase the risk of women's engagement in drug abuse and alcohol consumption [8]. Furthermore, physical IPV is linked to a high risk of mental health distress inkling post-traumatic stress disorders [9, 10]. Despite the strides Kenya has made in establishing legal and policy frameworks to address physical IPV [11], challenges persist due to limited coordination among sectors and service providers, inadequate financial and human resources, insufficient equipment, and a lack of knowledge among service providers [12]. Additionally, flawed evidence collection practices hinder enforcement and successful prosecution efforts [12]. Consequently, most cases of physical IPV remain underreported, primarily due to financial barriers and fear of encountering an unsupportive or discriminatory response from service providers [12]. Thus, making physical IPV an enduring social canker and public health concern.

Previous studies conducted in Kenya have shown that factors such as polygamous relationships, use of alcohol, history of childhood abuse and level of education predict intimate partner violence among women [13]. Moreover, age, place of residence, employment and ethnicity were found to predict intimate partner violence among women [14]. However, these studies did not utilise geospatial techniques in their analysis. Geospatial disparities can be considered a significant factor in explaining physical violence against women. Studies have shown that the geospatial technique contributes significantly to studying and proposing interventions or recommendations on health and social issues [15, 16]. This is especially critical as previous

evidence alludes to the fact that geospatial variation plays some role in the distribution of intimate partner violence [17].

A literature search revealed that studies have not been done to examine physical violence among women in Kenya using geospatial analysis. Resource constraints may impede the country's ability to focus specific interventions targeting the most vulnerable victims of physical violence in the country. Findings from this study will, therefore, assist policymakers and stakeholders in putting in place further policies as well as strengthening interventions aimed at addressing intimate physical violence among women in Kenya. This study aims to examine the spatial distribution and determinants of physical intimate violence among women in Kenya.

## Methods

### Data source

The data for this study were obtained from the 2022 Kenyan Demographic and Health Survey (KDHS), a nationally representative household survey designed to gather comprehensive data on population, health, and nutrition indicators. The Kenya National Bureau of Statistics conducted the survey in collaboration with the Ministry of Health, while ICF Macro provided technical assistance through the DHS program. The 2022 KDHS incorporates a two-stage sampling design. In the first stage, 1,692 clusters were randomly selected from the Kenya Household Master Sample Frame (K-HMSF) using equal probability with independent selection in each sampling stratum [18]. Subsequently, household listing was carried out in all selected clusters, creating a sampling frame for the second stage. In the second stage, 25 households were chosen from each cluster, resulting in 42,022 households being included in the survey sample. Data collection was performed using structured household and women's questionnaires administered by trained enumerators. The women's questionnaire covered various aspects, including birth history, childhood mortality, fertility preferences, child health, maternal health, and domestic violence. The analysis consisted of a weighted sample of 14,724 women between the ages of 15–49 years old having information on emotional violence considered in this study. The use of the dataset did not require ethical clearance since it was obtained from a secondary data source. Nevertheless, permission to use the dataset was obtained from the Measure DHS. This study relied on the Strengthening the Reporting of Observational Studies in Epidemiology (STROBE) guidelines in preparing this paper.

### Study variables and measurements

**Outcome variable.** The outcome variable for the study was lifetime experience physical violence. The response was captured as"yes" or "no".

**Explanatory variables.** Multiple factors informed by theoretical and empirical literature relating to physical violence [10, 14, 19] were included in the analyses as the explanatory variables. These variables include age (15–19, 20–24, 25–29, 30–34, 35–39, 40–44, 45–49), residence (rural and urban), level of education and partner's education (no education, basic, secondary and above), wealth index (poorest, poorer, middle, richer, richest), partner domineering variables include husband/partner are jealous if respondent talks with other men (never, often, sometimes, yes, but not in the last 12 months), husband/partner accuses respondent of unfaithfulness (never, often, sometimes, yes, but not in the last 12 months), husband/partner does not permit respondent to meet female friends (never, often, sometimes, yes, but not in the last 12 months), husband/partner tries to limit respondent's contact with family (never, often, sometimes, yes, but not in the last 12 months), husband/partner who insists on knowing where the respondent is (never, often, sometimes, yes, but not in the last 12 months).

Justification of wife beaten (no, yes) and Generational violence variables include Mother ever physically hurt (no, yes), father ever physically hurt (no, yes).

## Data analysis

Statistical analysis was performed using Stata version 17. Both descriptive and inferential statistics were conducted. Specifically, univariate and multivariate analyses were conducted to explore the relationships between the explanatory variables and outcome variables. Logistic regression models were fitted using Stata regression commands to assess adjusted risk factors associated with the study outcomes, with Odds Ratios (OR) and 95% confidence intervals (CI) calculated. To account for any sampling bias from under or over-sampling of respondents in the total population, the authors weighted all descriptive estimates using the individual weight variable (d005) in the dataset. In addition, the "svyset" command in Stata was employed to account for the complex survey design of the DHS data.

Spatial analysis was used in this study to visualise the distribution of physical violence among women in Kenya at the sub-county level. Respondent coordinates were obtained from the Measure DHS website and used in the analyses. The Kenya sub-county shapefile obtained from the OCHA service (https://data.humdata.org/dataset/cod-ab-ken?) was linked to the respondent's coordinates. The sub-county names were merged with the surveyed data to tie the sub-county information to the respondents sampled in the study. Essential variables were then extracted from the 2022 KDHS. The DHS cluster numbers and the coordinates from the retrieved non-spatial data (KDHS data) were combined using the one-to-many relational approach in SPSS version 25. As part of the data preparation, respondents who have experienced any form of physical violence (Yes) were assigned one (1), whereas those who have not experienced any form of physical violence (No) were assigned zero (0). All datasets (KDHS and sub-county shapefile) were projected to UTM zone 30s in ArcMap version 10.8. to determine the number of respondents and identify each case location, a spatial join was performed on the projected data to transfer the extracted data to the county level. Cluster aggregation was performed for sub-counties with more than one respondent. This process is done to derive the average values of women who have experienced any form of physical violence at the sub-county level.

The spatial autocorrelation (Global Moran's I) tool in ArcMap version 10.8 was used to analyse the spatial pattern of physical violence among women in Kenya. The spatial autocorrelation was done based on the hypothesis that physical violence perpetrated against women in Kenya is randomly distributed. The null hypothesis is rejected when the z-score is greater than ±1.65, implying that the observed spatial pattern is not likely to result from random events. Hotspot analysis (Getis-Ord G) was performed to determine the statistically significant spatial variation of physical violence against women in Kenya. The Hotspot helps to identify sub-county with significant high and low values of physical violence perpetrated against women in Kenya. Cluster and Outlier (Anselin Local Moran's I) analysis was performed to identify sub-county that appeared as outliers with their neighbouring sub-county.

To determine independent variables that explain the observed pattern, a Geographically Weighted Regression (GWR) analysis was conducted. The purpose of adopting this analytical tool is that it offers a robust statistical model in explaining how explanatory variables predict dependent variables. Prior to conducting the GWR analysis, a global OLS analysis was conducted to identify explanatory variables that predict physical violence among women in Kenya. It is worth stating that all explanatory variables stated in the statistical analysis were used in this analysis. The identified predictors were used in the GWR to determine the spatial predictors of physical violence among women in Kenya. The results of the spatial analyses are presented in figures and maps.

### Ethical consideration

This study did not seek ethical approval because our analysis relied on publicly accessible data. Nevertheless, the Demographic and Health Survey (DHS) documentation confirms that they obtained both written and verbal informed consent from all participants. Before commencing the survey, ethical clearance was obtained, and all ethical principles concerning the involvement of human subjects were rigorously followed. The methods employed were in full compliance with the guidelines and regulations outlined by the Declaration of Helsinki.

## Results

### Background characteristics

Table 1 displays the proportion of women who experienced physical intimate partner violence by background characteristics with corresponding chi-square ($X^2$) test scores. A total of 42,022 women were used for the study. Out of the total sample, 28.8% (10,477) reported having experienced physical intimate violence. The prevalence of experienced physical intimate violence was higher among women in their late reproductive age (34.1%) compared to young women aged 15–19 years (20.4%). The prevalence of physical intimate violence was higher in rural settings (33%) than in urban areas (22.1%). A smaller percentage of women with higher education (11.7%) had experienced physical intimate violence compared to 30.3% among women with no formal education. A similar pattern was observed among women from households with the highest wealth index (18%) compared to their counterparts from the poorest households (37%).

Among respondents with partners who had a higher educational level, a smaller number had experienced physical intimate violence (14.6%) compared to 30.1% and 38% among their counterparts whose partners have no formal education and primary education, respectively. Regarding the partner domineering indicators, lower proportions of women whose partners never feel jealous seeing them talk with other men (17%), never accuse them of unfaithfulness (21.8%), never prevent them from meeting their female friends (23.8), never try to limit them of their contact with family (25.1%), and never insist on knowing where they are (20.6%) had experienced physical intimate violence compared to their counterparts whose partners often or sometimes do otherwise. Using the chi-square test score, all the aforementioned independent variables show statistical difference across various categories.

### Spatial pattern of physical intimate partner violence among women in Kenya

The result from Moran's I spatial autocorrelation analysis in Fig 1 shows a z-score value of 2.58, indicating that physical intimate partner violence against women in Kenya did not occur randomly but rather clustered among some sub-countries in the country at a 99% confidence level. Moran's I spatial autocorrelation tool could not show the precise locations where clustering can be observed. The authors, to address the gap in Moran's tool, employed the Getis-Ord Gi hotspot analysis to visualise the spatial distribution of physical intimate partner violence against women in Kenya.

The hotspot analysis finds significant spatial clusters of high values (hot spots) and low values (cold spots) (Fig 2). It shows the intensity of the phenomena being studied. Moreover, it shows the spatial distribution of physical intimate partner violence against women in Kenya. Areas in red show a prevalence of physical intimate partner violence among women, while areas in blue also show a low occurrence of physical intimate partner violence against women. The map shows that women who reside in sub-counties in the southwest and central parts of

**Table 1.** Background characteristics, proportion experienced physical intimate partner violence.

| Background characteristics | Frequency N = 10,477 | Proportion experienced physical intimate partner violence | Chi-square($X^2$), p-value |
|---|---|---|---|
| **Demographic** | | | |
| *Age* | | | $X^2$ = 46.3, (<0.001) |
| 15–19 | 257 | 20.4 | |
| 20–24 | 1,570 | 25.3 | |
| 25–29 | 2,473 | 27.1 | |
| 30–34 | 2,067 | 27.3 | |
| 35–39 | 1,817 | 32.3 | |
| 40–44 | 1,308 | 31.7 | |
| 45–49 | 985 | 34.1 | |
| *Place of residence* | | | $X^2$ = 85.0 (0.002) |
| Urban | 3,966 | 22.1 | |
| Rural | 6,511 | 33.0 | |
| *Education* | | | $X^2$ = 314.0 (<0.001) |
| No education | 771 | 30.3 | |
| Primary | 4,331 | 37.1 | |
| Secondary | 3,398 | 28.0 | |
| Higher | 1,977 | 11.7 | |
| *Wealth status* | | | $X^2$ = 158.0 (<0.001) |
| Poorest | 1,823 | 37.0 | |
| Poorer | 1,879 | 34.9 | |
| Middle | 1,985 | 32.8 | |
| Richer | 2,296 | 25.7 | |
| Richest | 2,494 | 18.0 | |
| **Marital status** | | | $X^2$ = 4.3 (0.040) |
| Married | 9,107 | 28.6 | |
| Cohabitation | 1,370 | 30.6 | |
| **Occupation** | | | $X^2$ = 164.8 (<0.001) |
| Not working | 3,498 | 26.0 | |
| Professional/technical | 1,839 | 19.2 | |
| Clerical | 141 | 23.7 | |
| Sales | 819 | 27.3 | |
| Agriculture–self-employed | 62 | 26.9 | |
| Agriculture—employee | 1,963 | 36.7 | |
| Household and domestic | 487 | 38.9 | |
| Services | 576 | 35.1 | |
| Skilled manual | 124 | 29.5 | |
| Unskilled manual | 826 | 37.3 | |
| Don't know | 142 | 20.0 | |
| **Partner's education** | | | $X^2$ = 248.6 (<0.001) |
| No education | 640 | 30.1 | |
| Primary | 4,031 | 38.1 | |
| Secondary | 3,403 | 27.7 | |
| Higher | 2,403 | 14.6 | |
| **Partner Domineering** | | | |
| *Husband/partner jealous if respondent talks with other men* | | | $X^2$ = 813.5 (<0.001) |
| Never | 5,460 | 17.0 | |

*(Continued)*

**Table 1.** (Continued)

| Background characteristics | Frequency N = 10,477 | Proportion experienced physical intimate partner violence | Chi-square($X^2$), p-value |
|---|---|---|---|
| Often | 1,565 | 48.6 | |
| Sometimes | 2,767 | 38.5 | |
| Yes, but not in the last 12 months | 685 | 39.4 | |
| *Husband/partner accuses respondent of unfaithfulness* | | | $X^2$ = 1.1e+03 (<0.001) |
| Never | 8,646 | 21.8 | |
| Often | 419 | 71.9 | |
| Sometimes | 1.096 | 59.4 | |
| Yes, but not in the last 12 months | 316 | 59.1 | |
| *Husband/partner does not permit respondent to meet female friends* | | | $X^2$ = 786.1 (<0.001) |
| Never | 9,083 | 23.8 | |
| Often | 482 | 64.7 | |
| Sometimes | 800 | 57.0 | |
| Yes, but not in the last 12 months | 112 | 78.5 | |
| *Husband/partner tries to limit respondent's contact with family* | | | $X^2$ = 702.4 (<0.001) |
| Never | 9,623 | 25.1 | |
| Often | 272 | 71.5 | |
| Sometimes | 484 | 68.9 | |
| Yes, but not in the last 12 months | 98 | 78.9 | |
| *Husband/partner insists on knowing where respondent is* | | | $X^2$ = 762.3 (<0.001) |
| Never | 7,298 | 20.6 | |
| Often | 1,630 | 48.85 | |
| Sometimes | 1,439 | 46.1 | |
| Yes, but not in the last 12 months | 110 | 56.7 | |
| **Justification of wife beaten** | | | $X^2$ = 182.7 (<0.001) |
| No | 6,966 | 23.4 | |
| Yes | 3,511 | 39.6 | |
| **Generational violence** | | | |
| *Mother ever physically hurt* | | | $X^2$ = 8.2 (0.004) |
| No | 10,215 | 28.8 | |
| Yes | 262 | 31.1 | |
| *Father ever physically hurt* | | | $X^2$ = 2.5 (0.117) |
| No | 10,305 | 28.8 | |
| Yes | 172 | 31.2 | |
| **Total** | **10,477** | **28.8** | |

Kenya experience more physical abuse. Sub -counties such as Chepaluugu, Konion, Sotik, Bumula, Kabuchai, Bomet Central, Metayos, Teso North, Teso South, Ndhiwa, Suba, Butere, Mumias East, Mumais West, Bonchari, Seme, South Mugirango, Kisumu West, Kisumu Central, Muhoroni Nyatike, Awendo among other. Respondents in these areas responded that they had experienced one or more forms of physical intimate partner violence.

On the other hand, areas in blue, as shown in the map (Fig 2) in the North East corner and the South East corner, were indicated as cold spots. Sub-counties such as Kisauni, Tarabaj, Waijir North, Lafey, Mandera North and South, Mandera West, Banissa, Lungalungo, Kinango, Matuga, Msambweni, Malindi, Rabi, Kilifi North and South, Ganze among others

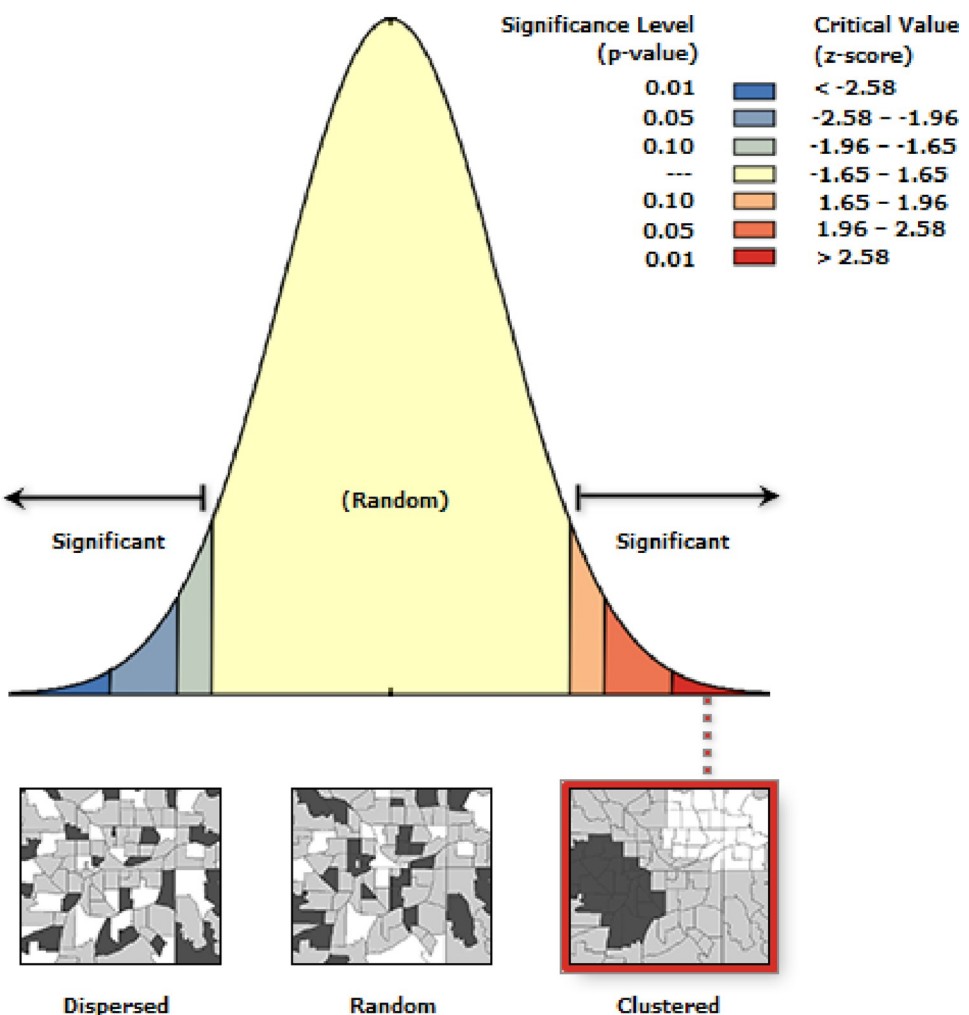

**Fig 1. Spatial distribution of physical intimate partner violence among women in Kenya. Source**: Authors' construct based on the 2022 KDHS data.

were areas that had women with little or no physical intimate partner violence experience compared to women that reside in the hotspot zone. Areas in light blue are sub-counties with a 95% confidence level of little or no experience of physical intimate partner violence against women.

Although the hotspot analysis gives a spatial visualisation of the sub-counties with high and low incidences of physical violence, the cluster and outlier analysis, as shown in Fig 3, revealed some unique findings generalised by the hotspot analysis. The map (Fig 3) showed that some sub-counties with high physical intimate partner violence cases (hotspots) were surrounded by sub-counties with low physical intimate partner violence cases (cold spots), and sub-counties with high physical intimate partner violence cases surrounded some sub-counties with low cases. The sub-counties with low physical intimate partner violence are shown in blue. Sub-counties include Suba, Budalangi, Samburu North, Alego, Bondo and Gem, which are cold spot zones surrounded by hotspot zones.

On the contrary, sub-counties such as Mbooni, Kibwezi East, and Kaiti were identified as hotspots (Fig 3). Thus, women who reside in these sub-counties are prone to one or more

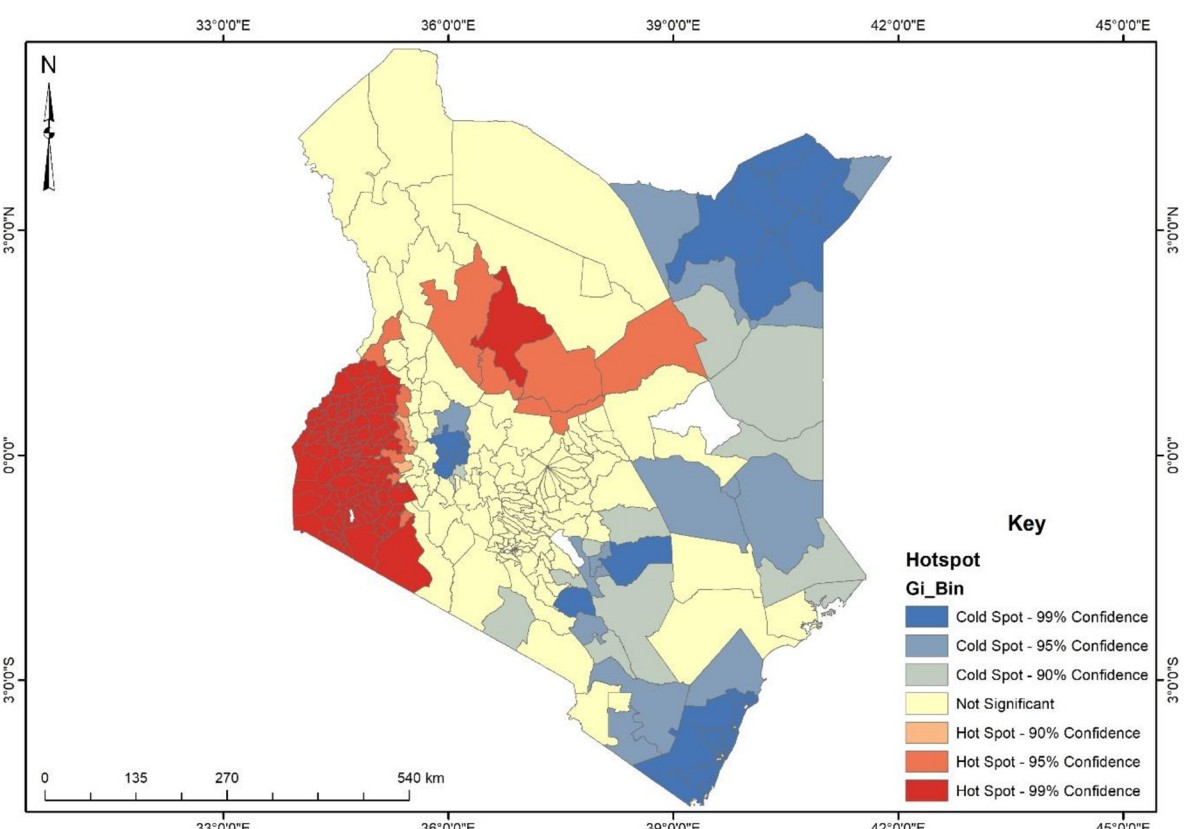

**Fig 2. Hotspot analysis of physical intimate partner violence among women in Kenya. Source**: Authors' construct based on the 2022 KDHS data.

forms of physical intimate partner violence but are surrounded by sub-counties with little or no experience of physical intimate partner violence against women.

## Multivariable logistic regression results

In Table 2, we used multivariable logistic regression analysis to identify factors associated with the experience of physical intimate partner violence among reproductive-age women (15–49 years) in Kenya. The results were estimated in adjusted odds ratio with corresponding significance levels between the independent variables and the outcome of interest. Our findings revealed that age, education, wealth index, type of occupation, partner domineering indicators, and justification of wife beating were significantly associated with the experience of physical intimate partner violence among the studied population. The findings show an increased odds of experiencing physical intimate partner violence with an increase in age. For instance, compared with young women aged 15–19 years, those aged 45–49 years were 3.13 times higher in experiencing physical intimate partner violence. Improvement in women's educational level and wealth status served as a protective factor against physical intimate partner violence. Thus, women with higher levels of formal education were less likely to experience physical intimate partner violence [aOR = 0.54, CI = 0.36–0.81] compared to those who had no formal education. A similar trend was observed among women in the richer [aOR = 0.75, CI = 0.58–0.95] and richest wealth index [aOR = 0.72, CI = 0.52–0.99] compared to their counterparts from the poorest household.

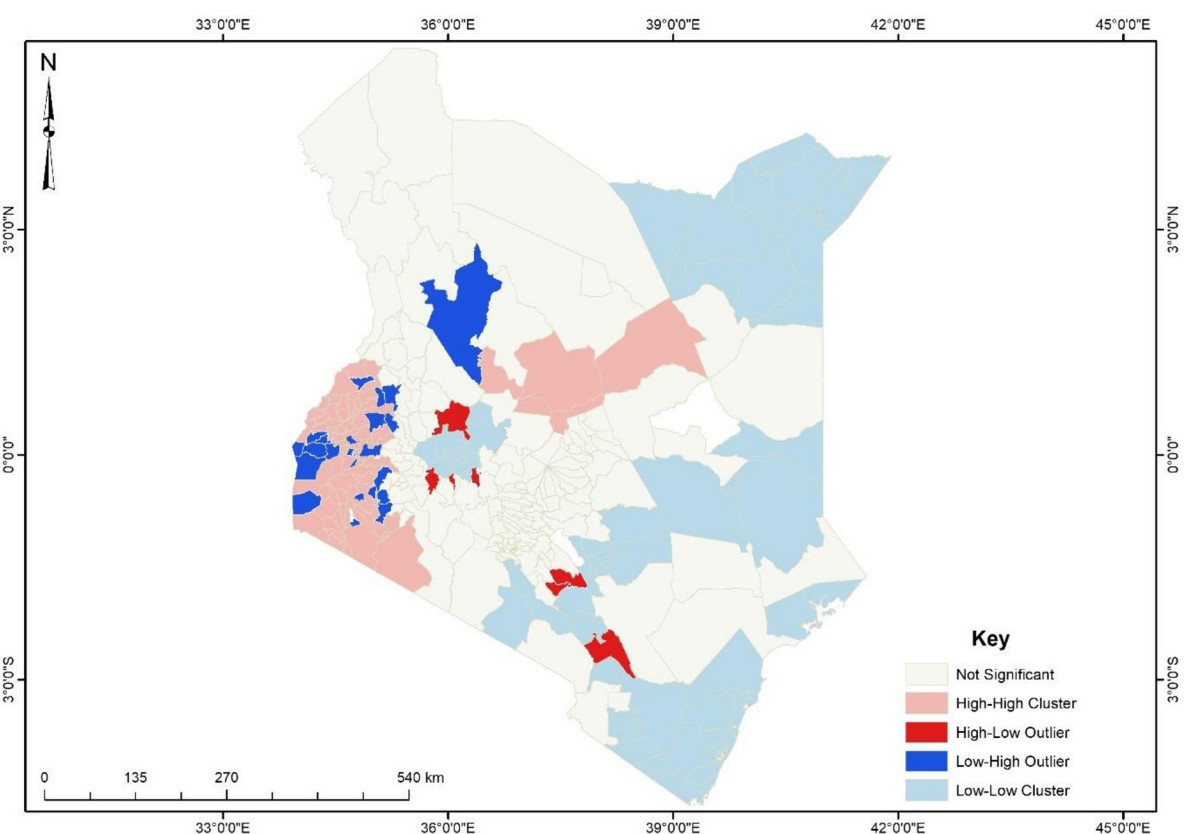

**Fig 3. Cluster and outlier analysis physical intimate partner violence among women in Kenya. Source**: Authors' construct based on the 2022 KDHS data.

Conversely, higher odds of physical intimate partner violence were found among women who often or sometimes experience various indicators of partner domineering than those who never experience them. For example, women whose partners often or sometimes feel jealous seeing them talk to other men were 1.94 and 1.87 times more likely to experience physical intimate partner violence than those whose partners never feel jealous seeing them talk to other men. Relatedly, women who justify of wife being beaten had a higher likelihood of experiencing physical intimate partner violence [aOR = 1.62, CI = 1.39–1.88] compared to their counterparts who do not endorse it (see Table 2).

## Spatial predictors of physical violence among women in Kenya

The Ordinary Least Square (OLS) analysis revealed that all explanatory variables explained about 44% ($R^2$ = 0.439) of the observed spatial pattern and the F-Statistics was statistically significant at 99% confidence indicating that there is a good overall model fit (Fig 4). It is worth stating that two explanatory variables were found to be statistically significant and these were wealth status (richest) and Husband/partner jealous if respondent talks with other men (Sometimes).

From the GWR, the two explanatory variables explained about 53% ($R^2$ = 0.527) of the spatial pattern of physical violence among women in Kenya. With regards to wealth status (richest), it was found that the explanatory variable had both positive and negative predictive power at the sub-counties level. Evidence from Fig 4 shows that most women in the north (North

**Table 2. Bivariate analysis of experienced physical intimate partner violence.**

| Background characteristics | Adjusted Odds Ratio (AOR) | Confidence interval (CI) |
|---|---|---|
| **Demographic** | | |
| *Age* | | |
| 15–19 | Ref | Ref |
| 20–24 | 1.68 | 0.99, 2.85 |
| 25–29 | 2.33** | 1.41, 3.86 |
| 30–34 | 2.20*** | 1.32, 3.67 |
| 35–39 | 2.63*** | 1.57, 4.41 |
| 40–44 | 2.74*** | 1.62, 4.63 |
| 45–49 | 3.13*** | 1.82, 5.36 |
| *Place of residence* | | |
| Urban | Ref | Ref |
| Rural | 1.17 | 0.94, 1.46 |
| *Education* | | |
| No education | Ref | Ref |
| Primary | 1.14 | 0.88, 1.48 |
| Secondary | 1.05 | 0.78, 1.42 |
| Higher | 0.54** | 0.36, 0.81 |
| *Wealth status* | | |
| Poorest | Ref | Ref |
| Poorer | 0.92 | 0.75, 1.12 |
| Middle | 0.83 | 0.68, 1.02 |
| Richer | 0.75** | 0.58, 0.95 |
| Richest | 0.72* | 0.52, 0.99 |
| **Marital status** | | |
| Married | Ref | Ref |
| Cohabitation | 1.01 | 0.83, 1.24 |
| **Occupation** | | |
| Not working | Ref | Ref |
| Professional/technical | 0.93 | 0.75, 1.17 |
| Clerical | 1.27 | 0.58, 2.78 |
| Sales | 1.22 | 0.90, 1.65 |
| Agriculture–self employed | 0.98 | 0.49, 1.94 |
| Agriculture—employee | 1.07 | 0.89, 1.29 |
| Household and domestic | 1.41* | 1.02, 1.94 |
| Services | 1.70** | 1.23, 2.35 |
| Skilled manual | 1.21 | 0.75, 1.96 |
| Unskilled manual | 1.51** | 1.16, 1.96 |
| Don't know | 1.25 | 0.65, 2.40 |
| **Partner's education** | | |
| No education | Ref | Ref |
| Primary | 1.29 | 0.96, 1.73 |
| Secondary | 1.04 | 0.75, 1.44 |
| Higher | 0.82 | 0.55, 1.20 |
| **Partner Domineering** | | |
| *Husband/partner jealous if respondent talks with other men* | | |
| Never | Ref | Ref |

*(Continued)*

**Table 2.** (Continued)

| Background characteristics | Adjusted Odds Ratio (AOR) | Confidence interval (CI) |
|---|---|---|
| Often | 1.94*** | 1.57, 2.40 |
| Sometimes | 1.87*** | 1.60, 2.19 |
| Yes, but not in the last 12 months | 1.69*** | 1.27, 2.25 |
| *Husband/partner accuses respondent of unfaithfulness* | | |
| Never | Ref | Ref |
| Often | 2.69*** | 1.91, 3.80 |
| Sometimes | 2.60*** | 2.10, 3.21 |
| Yes, but not in the last 12 months | 2.90*** | 2.02, 4.17 |
| *Husband/partner does not permit respondent to meet female females* | | |
| Never | Ref | Ref |
| Often | 1.75** | 1.28, 2.39 |
| Sometimes | 1.54** | 1.18, 2.02 |
| Yes, but not in the last 12 months | 4.39*** | 2.30, 8.38 |
| *Husband/partner tries to limit respondent's contact with family* | | |
| Never | Ref | Ref |
| Often | 2.37*** | 1.53, 3.67 |
| Sometimes | 2.79*** | 2.02, 3.86 |
| Yes, but not in the last 12 months | 4.40*** | 2.09, 9.25 |
| *Husband/partner insists on knowing where respondent is* | | |
| Never | Ref | Ref |
| Often | 1.62*** | 1.33, 1.97 |
| Sometimes | 1.79*** | 1.49, 2.15 |
| Yes, but not in the last 12 months | 1.17 | 0.56, 2.43 |
| **Justification of wife beaten** | | |
| No | Ref | Ref |
| Yes | 1.62*** | 1.39, 1.88 |
| **Generational violence** | | |
| *Mother ever physically hurt* | | |
| No | Ref | Ref |
| Yes | 1.21 | 0.77, 1.90 |
| *Father ever physically hurt* | | |
| No | Ref | Ref |
| Yes | 1.02 | 0.57, 1.20 |

*p<0.05

**p<0.01

***p<0.001 Ref = Reference category

Horr, Laisamis, Turkana Central, etc.) were less likely (negative coefficients) to experience physical violence because a unit increase in women's wealth status to richest accounts for 2–3% decrease in physical violence. This was general for most of the places, however, it was found that four sub-counties (Mandera West, Mandera South, Tarbaj and Wajir East) had a positive coefficient which suggests that a unit increase in women's wealth status to richest account for about 1–6% increase in physical violence (Fig 4).

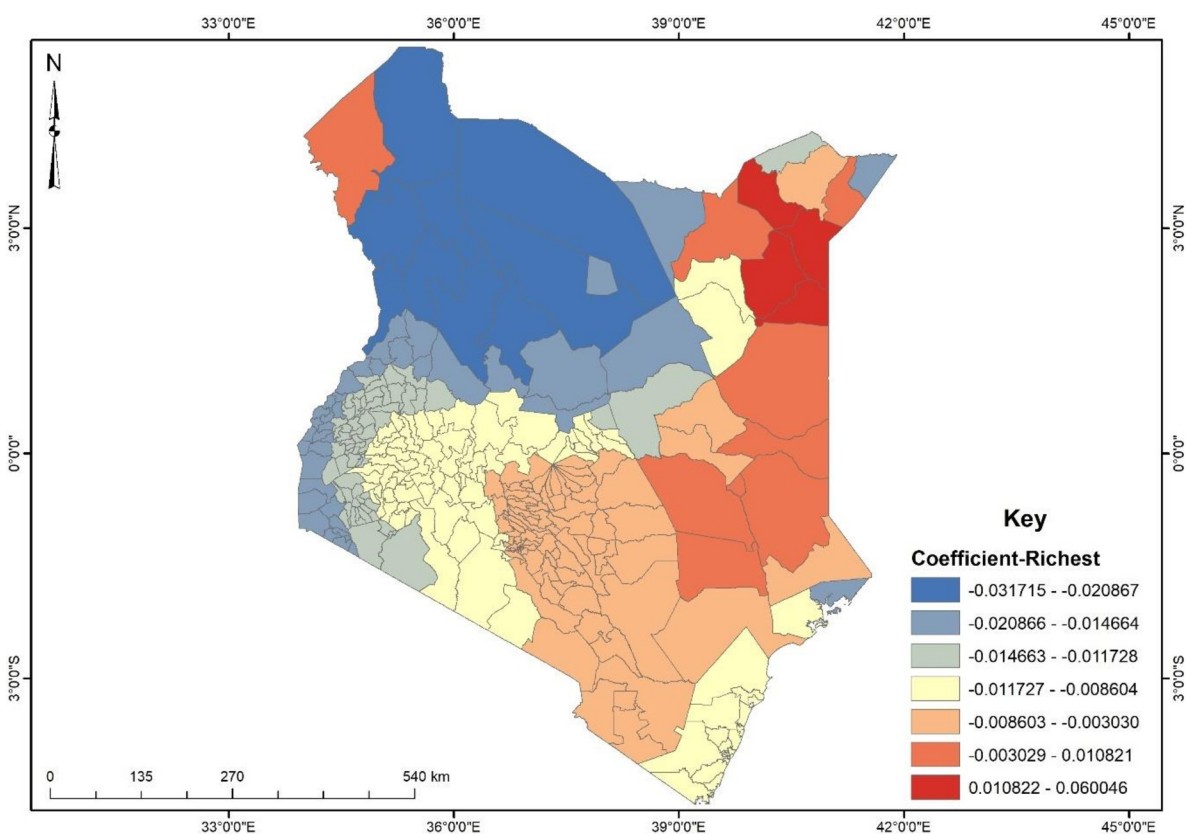

**Fig 4. GWR coefficient of wealth status (richest). Source**: Authors' construct based on the 2022 KDHS data.

Evidence from Fig 5 shows that some sub-counties had a unit increase in Husband/partner jealous if respondent talks with other men (Sometimes) accounting for about 0.6–2% decrease in physical violence among Kenyan women. This was evident in the northwest and eastern parts of Kenya including sub-counties such as Turkana West, Loima, Wajir South and Dadaab. It can be seen that most of the sub-counties' coefficients were positive and ranged between 0.002–0.018. This suggests that a unit increase in Husband/partner jealous if respondent talks with other men (Sometimes) accounts for a 0.2–1.8% increase in physical violence among Kenyan women.

## Discussion

The study aimed to assess the geospatial disparities in physical intimate violence among women in Kenya. Our study revealed that 28.8% of Kenyan women had experienced physical intimate violence. The prevalence of physical intimate violence observed in our study exceeds the 19.7% reported in SSA [20], yet falls below previous findings in Kenya (35%) [6] and among Zambian women (44.7%) [21]. The disparity between our estimated prevalence and that of Kimuna et al. [6] suggests potential advancements in women's assertiveness over time. Notably, Kimuna et al.'s study [6] was based on the 2014 KDHS, whereas ours reflects data from the 2022 KDHS. It is plausible that improvements in women's ability to resist physical violence from their intimate partners have occurred during this interim period. Such progress may signify evolving societal attitudes and increased awareness surrounding intimate partner violence. Suppose nothing is done to reduce physical intimate partner violence, Kenya will be

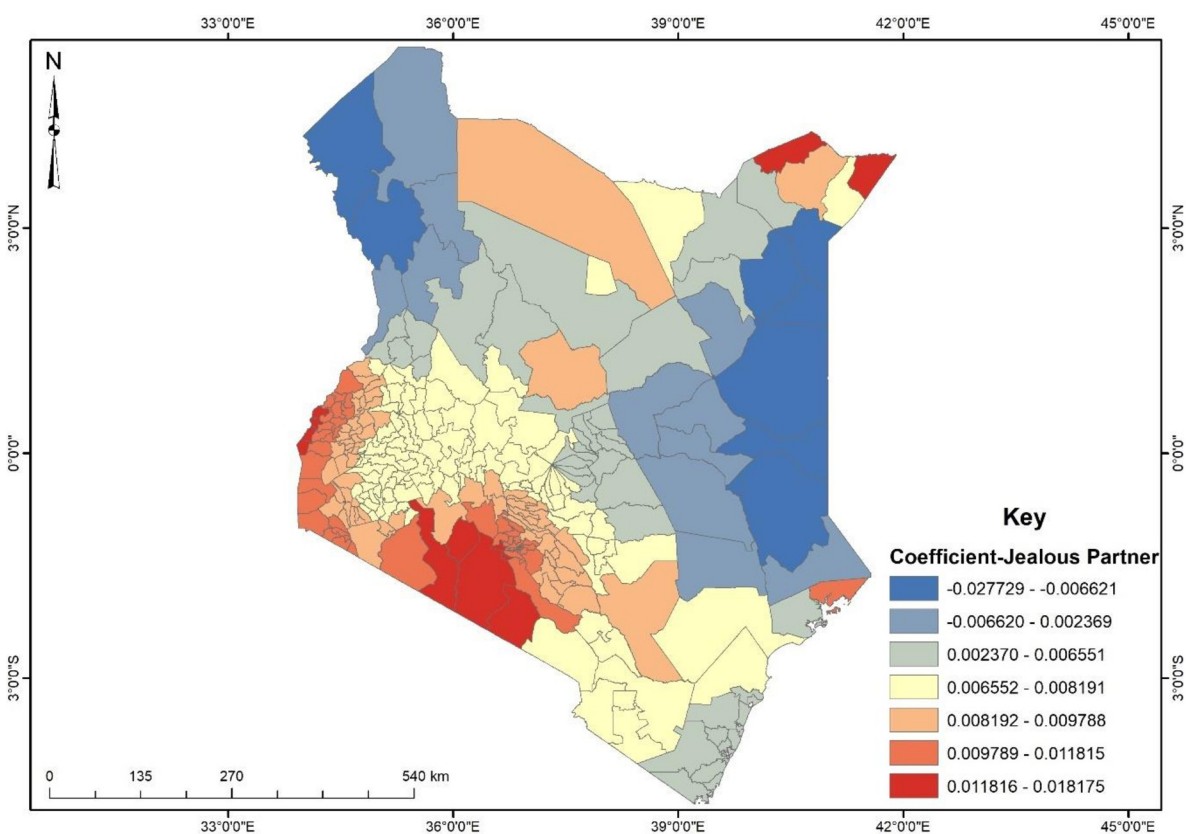

**Fig 5. GWR coefficient of husband/partner jealous if respondent talks with other men (sometimes). Source**: Authors' construct based on the 2022 KDHS data.

unable to achieve the Sustainable Development Goal (SDG) target 5.2.1, which seeks to eliminate all forms of violence by 2030 [22].

Our findings indicate that physical violence was not even across all geographic spaces in Kenya. There were some hotspots of physical intimate violence. This finding indicates that the phenomenon is clustered around certain areas of the country. The clustered nature of the physical intimate partner violence could be attributed to socio-cultural factors such as marital rites, poverty, low level of education, women's endorsement of wife beating and kinship [23]. For instance, the study by Alesina, Brioschi and La Ferrara [23] revealed that socio-cultural factors such as marital rites may expose women to physical intimate partner violence, but women caught up in endogamy and co-residing with their husband's family are more likely to experience violence.

Evidence from the present study shows that there is a positive association between age and the likelihood of women experiencing physical intimate partner violence. This finding is inconsistent with a study conducted in Uganda [24] that found no significant association between age and physical intimate partner violence. However, our result is corroborated by a study [25] that found significantly higher odds of physical intimate partner violence as age increases. One school of thought is that older women tend to document their lifetime accumulation of physical intimate partner violence experiences, as they have had more time during their lives to encounter instances of IPV compared to younger women [25]. Another plausible explanation could be that younger women may lack the assertiveness to accept and report acts of violence. Hence, they are likely to be underreported.

We found an inverse association between education and women's odds of experiencing physical intimate partner violence. This implies that women of higher educational attainment are significantly less likely to experience physical violence compared to those who have had no formal education. While the observed association is in contrast to Tiruye et al.'s [25] study, which found no significant association, our result resonates with a plethora of studies conducted in SSA [19] and Peru [26]. Women with higher levels of education often have better access to resources, information, and opportunities. This can empower them to make choices such as seeking support from social networks, accessing legal assistance, or pursuing economic independence, hence reducing their vulnerability to physical violence from an intimate partner.

Corollary to education, our study revealed that women with a higher wealth index were less likely to experience physical violence from an intimate partner compared to those with the poorest wealth index–a result that mirrors the findings from previous studies [19, 27]. The observed association is not surprising because at the heart of intimate partner violence is the issue of power dynamics. Particularly in SSA countries like Kenya, the partner with the most access to financial resources makes all decisions. Therefore, women in wealthier households often have better access to support systems, including friends, family, and professional networks. These networks can provide emotional and practical advice, making them less tolerant of violence and more likely to exit relationships that show tendencies for physical violence. Meanwhile, women in the poorest wealth index would be more tolerant of physical violence and find justifications [28].

Consistent with previous literature [29, 30], we found that women who endorse violence are more likely to experience physical violence from their intimate partner. This finding is expected because women who justify violence or accept that their partners are right to act violently in certain scenarios are more likely to be tolerant of intimate partner violence. Moreover, they are more likely to succumb to socio-cultural norms that reward subservient behaviours. Hence, partners may capitalise on this subservient behaviour and perpetuate physical violence, compared to women who show clear unacceptance of violence. Relatedly, our study revealed that women whose partners exhibited domineering attitudes were more likely to experience physical violence. A possible explanation is that partner domineering attitudes tend to overshadow the empowerment of women, as evidenced in a previous study which showed that women who exert a higher degree of household decision-making autonomy still experience a higher risk of intimate partner violence when they had partners who endorsed domineering attitudes.

### Strengths and limitations

In this study, we used the 2022 Kenya Demographic and Health Survey data. This data is current and, therefore, makes our findings reflect the current status quo of women in the country. Also, the large sample size gives us the statistical power to extrapolate our findings to women of reproductive age in Kenya. However, the data used was self-reported. Hence, there is the possibility of recall bias and social desirability bias. Although the data was collected at the peak of the COVID-19 pandemic, it does not contain any variables that could have helped to assess the dynamics of pandemics in predicting physical intimate partner violence. Also, the cross-sectional nature of the data precludes us from establishing causality.

### Conclusion

We conclude that there are spatial differences in women's risk of experiencing physical violence from an intimate partner. It is, therefore, imperative for the Kenyan government to roll

out policies and programmes that will target the hotspot areas of physical intimate partner violence. Such programmes must prioritise women in the poorest wealth index, those without formal education and older women. Also, social behavioural change communication campaigns must be implemented to reorient the population about the dangers of violence justification and partner domineering behaviours. What this suggests is that male partners must be actively involved in all programmes and initiatives that aim to eliminate intimate partner violence.

## Supporting information

**S1 Checklist. STROBE statement__physical intimate partner violence among women in Kenya.**
(DOCX)

## Acknowledgments

We are grateful to the DHS Program for providing us with access to the dataset.

## Author Contributions

**Conceptualization:** Joshua Okyere, Castro Ayebeng, Kwamena Sekyi Dickson.

**Data curation:** Kwamena Sekyi Dickson.

**Formal analysis:** Ebenezer N. K. Boateng, Bright Ankomahene, Kwamena Sekyi Dickson.

**Methodology:** Kwamena Sekyi Dickson.

**Writing – original draft:** Joshua Okyere, Castro Ayebeng, Ebenezer N. K. Boateng, Rebecca A. A. Assie, Amanda Odoi, King-David Dzirassah, Bright Ankomahene, Kwamena Sekyi Dickson.

**Writing – review & editing:** Castro Ayebeng, Ebenezer N. K. Boateng, Rebecca A. A. Assie, Amanda Odoi, Kwamena Sekyi Dickson.

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
