## [Decision Letter · Decision Letter 0]

25 Mar 2024

PONE-D-24-02046Spatial distribution and determinants of Physical Intimate Partner Violence among women in Kenya: Evidence from the 2022 Kenya Demographic and Health SurveyPLOS ONE

Dear Dr. Ayebeng,

Thank you for submitting your manuscript to PLOS ONE. After careful consideration, we feel that it has merit but does not fully meet PLOS ONE’s publication criteria as it currently stands. Therefore, we invite you to submit a revised version of the manuscript that addresses the points raised during the review process.

**ACADEMIC EDITOR:**Please, carefully review all the Reviewers' comments and respond appropriately. Where you disagree, provide point-by-point explanation of why you disagree with their comments.=============================

We look forward to receiving your revised manuscript.

Kind regards,

Adobea Yaa Owusu, MA, PhD, MPH

Academic Editor

PLOS ONE

Journal Requirements:

3. We note that [Figures 2 and 3] in your submission contain [map/satellite] images which may be copyrighted. All PLOS content is published under the Creative Commons Attribution License (CC BY 4.0), which means that the manuscript, images, and Supporting Information files will be freely available online, and any third party is permitted to access, download, copy, distribute, and use these materials in any way, even commercially, with proper attribution. For these reasons, we cannot publish previously copyrighted maps or satellite images created using proprietary data, such as Google software (Google Maps, Street View, and Earth). For more information, see our copyright guidelines: http://journals.plos.org/plosone/s/licenses-and-copyright.

a. You may seek permission from the original copyright holder of Figures 2 and 3 to publish the content specifically under the CC BY 4.0 license.  

Reviewers' comments:

Reviewer's Responses to Questions

**Comments to the Author**

1. Is the manuscript technically sound, and do the data support the conclusions?

Reviewer #1: Yes

Reviewer #2: Yes

2. Has the statistical analysis been performed appropriately and rigorously? 

Reviewer #1: Yes

Reviewer #2: Yes

3. Have the authors made all data underlying the findings in their manuscript fully available?

Reviewer #1: Yes

Reviewer #2: Yes

4. Is the manuscript presented in an intelligible fashion and written in standard English?

Reviewer #1: Yes

Reviewer #2: Yes

5. Review Comments to the Author

Reviewer #1: I congratulate the authors on the manuscript. Overall, it is well-written and publishable. The idea of spatial variation is an interesting one. Indeed, it is wrong to assume that a unified approach in a country, especially in Africa where cultural and gender norms vary. Hence, attitude towards IPV and the experience of it may vary across locations.

Overall, I have two general comments. One, the spatial analysis employed by the authors is merely descriptive. It only shows that physical IPV is higher in one place than another. But is that enough? How does the influence of those factors vary across different regions? The author may be surprised to find that a factor that is significant in one location is insignificant in another. In fact, the authors may find that a particular variable predicts IPV in opposite directions in two different regions. Please see the work of my colleague, Ramsden in Nigeria. https://doi.org/10.1017/S0021932022000463 The authors may want to group different locations into two, three, or four and see how the influence of the factors varies across regions. This is important for policy implementation.

Secondly, the authors, like many previous authors, assume a uni-directional association between IPV justification and IPV experience. The authors mentioned this a few times from the abstract, results and discussion section. For example, the authors stated, “…we found that women who endorse violence are more likely to experience physical violence from their intimate partner. This finding is expected because women who justify violence or accept that their partners are right to act violently in certain scenarios are more likely to be tolerant of intimate partner violence.” I doubt if this is the case with DHS data. I can argue convincingly in the opposite direction. If a woman has experienced IPV, for example, she is more likely to justify it than a woman who has never had experience. It is common for human beings to justify things they do or have experienced. To the best of my knowledge and what I read in the work of Ramsden in Nigeria, the association is bi-directional. The authors may want to further contribute to the knowledge of IPV in Kenya by running an analysis using IPV experience as the independent variable and IPV justification as the dependent.

In the introduction, the authors should explore the DHS and state the proportion of women who have experienced different forms of IPV: physical, emotional and sexual. I predict that the rate of physical IPV will be higher than emotional and sexual. This will allow the paper to establish physical IPV as a major problem that requires attention.

To my surprise, the authors were also silent on the explanations for regional differences in the experience of IPV. Only the second paragraph of the introduction was dedicated to discussing the variations, but no explanation was provided based on the knowledge of the regions in Kenya. Based on the authors’ knowledge of the regions, what could explain the variations? Is any of the seven authors from Kenya? I see that all of them are based in Ghana.

Please do some proofreading of the manuscript for typo and grammar errors. You wrote “marita status” in the table on page 21.

Reviewer #2: Thank you for the opportunity to review this manuscript. The paper is worthwhile – especially considering the persistence of IPV in sub-Saharan Africa. Below I note some comments and concerns that will help improve the paper.

Introduction

The introduction does not provide a compelling case for why the study is important, particularly within the Kenyan context. Why is Physical IPV a serious issue in Kenya? What makes it endure? What are the effects of this form of IPV, etc.? Also, justify the focus on physical violence and not the other forms.

The last two sentences of the first paragraph - “Globally, approximately 30 per cent of women have been exposed to intimate physical violence. While in Kenya, approximately 35 per cent of women experience physical violence orchestrated by their intimate partners”. The second sentence is incomplete. I suggest putting a comma between the two sentences might make the comparison clearer.

Methods

Outcome variable- please specify if this is lifetime violence or just in the past twelve months.

Results

Please comment on some of the wide CIs e.g., age 44-491.82, 5.36

Discussion

The study found that 28.8% of participants had experienced physical IPV. In the introduction, the authors noted that previous studies reported an average of 35%. It is important to comment on this disparity. Could it be different data sets or time frames- What does this reduction mean?

A significant finding from the study is the observed geographical variation. What explanations could account for this variation, and what implications does it carry?

6. PLOS authors have the option to publish the peer review history of their article (what does this mean?). If published, this will include your full peer review and any attached files.

Reviewer #1: No

Reviewer #2: No

---

## [Author Response · Author response to Decision Letter 0]

4 Jun 2024

PONE-D-24-02046

Re: “Spatial distribution and determinants of Physical Intimate Partner Violence among women in Kenya: Evidence from the 2022 Kenya Demographic and Health Survey”

Dear Prof. Adobea Yaa Owusu,

We are grateful to you and the reviewers for your comments on our paper entitled " Spatial distribution and determinants of Physical Intimate Partner Violence among women in Kenya: Evidence from the 2022 Kenya Demographic and Health Survey". We would also take this opportunity to thank the reviewers for finding merit in this paper and suggesting some revisions. We have taken notice of all the comments raised by the reviewers and have responded accordingly as follows. Please note that the reviewers' comments are in black whereas our responses are in red. 

Journal Requirements:

Response: We have ensured this, thank you. 

Response: We have ensured this.

3. We note that [Figures 2 and 3] in your submission contain [map/satellite] images which may be copyrighted. All PLOS content is published under the Creative Commons Attribution License (CC BY 4.0), which means that the manuscript, images, and Supporting Information files will be freely available online, and any third party is permitted to access, download, copy, distribute, and use these materials in any way, even commercially, with proper attribution. For these reasons, we cannot publish previously copyrighted maps or satellite images created using proprietary data, such as Google software (Google Maps, Street View, and Earth). For more information, see our copyright guidelines: http://journals.plos.org/plosone/s/licenses-and-copyright.

Response: These are original maps designed from this study’s analyses.

a. You may seek permission from the original copyright holder of Figures 2 and 3 to publish the content specifically under the CC BY 4.0 license. 

Response: These are original maps designed from this study’s analyses.

Response: These are original maps designed from this study’s analyses.

Reviewers' comments:

Reviewer #1:

 I congratulate the authors on the manuscript. Overall, it is well-written and publishable. The idea of spatial variation is an interesting one. Indeed, it is wrong to assume that a unified approach in a country, especially in Africa where cultural and gender norms vary. Hence, attitude towards IPV and the experience of it may vary across locations.

Response: Thank you

Overall, I have two general comments. One, the spatial analysis employed by the authors is merely descriptive. It only shows that physical IPV is higher in one place than another. But is that enough? 

Response: Thanks for your comment. A geographically weighted regression (GWR) analysis has be included to identify predictors of the observed spatial pattern.

How does the influence of those factors vary across different regions? 

Response: It has been presented in the GWR results

The author may be surprised to find that a factor that is significant in one location is insignificant in another. In fact, the authors may find that a particular variable predicts IPV in opposite directions in two different regions. Please see the work of my colleague, Ramsden in Nigeria. https://doi.org/10.1017/S0021932022000463 The authors may want to group different locations into two, three, or four and see how the influence of the factors varies across regions. This is important for policy implementation.

Response: Thank you for your concern but we considered using a spatial regression model (GWR) to predict the spatial variability of the observed pattern based on significant explanatory variables.

Secondly, the authors, like many previous authors, assume a uni-directional association between IPV justification and IPV experience. The authors mentioned this a few times from the abstract, results and discussion section. For example, the authors stated, “…we found that women who endorse violence are more likely to experience physical violence from their intimate partner. This finding is expected because women who justify violence or accept that their partners are right to act violently in certain scenarios are more likely to be tolerant of intimate partner violence.” I doubt if this is the case with DHS data. I can argue convincingly in the opposite direction. If a woman has experienced IPV, for example, she is more likely to justify it than a woman who has never had experience. It is common for human beings to justify things they do or have experienced. To the best of my knowledge and what I read in the work of Ramsden in Nigeria, the association is bi-directional. The authors may want to further contribute to the knowledge of IPV in Kenya by running an analysis using IPV experience as the independent variable and IPV justification as the dependent.

Response: Thank you for this invaluable contribution to the study. We appreciate the thoughtful consideration of alternative perspectives on the association between intimate partner violence (IPV) justification and IPV experience. While we acknowledge the validity of your viewpoint, we would like to highlight that our study's direction is also supported by existing literature and empirical evidence (https://link.springer.com/article/10.1186/s12905-022-01656-7;
https://www.tandfonline.com/doi/abs/10.1080/13691058.2020.1743880 ). 

Thus, over time women who justify IPV may develop tolerant attitudes toward IPV violence against women and consider the violence as normal in their life process.

In the introduction, the authors should explore the DHS and state the proportion of women who have experienced different forms of IPV: physical, emotional and sexual. I predict that the rate of physical IPV will be higher than emotional and sexual. This will allow the paper to establish physical IPV as a major problem that requires attention.

Response: Thank you. We have now provided this this information. It reads: “Evidence from 26 sub-Saharan African countries identifies physical violence (30.58%) as the major IPV suffered by women followed by emotional (30.22%) and sexual IPV (12.26) [5]. In the context of Kenya, approximately 35 percent of women experience physical violence orchestrated by their intimate partners [6].”

To my surprise, the authors were also silent on the explanations for regional differences in the experience of IPV. Only the second paragraph of the introduction was dedicated to discussing the variations, but no explanation was provided based on the knowledge of the regions in Kenya. Based on the authors’ knowledge of the regions, what could explain the variations? 

Response: Thank you very much for your comment. The plausible cause of the regional differences has been presented in the second paragraph of the discussion. In addition, a spatial regression model is conducted to determine factors that explain the observed spatial pattern of IPV.

Is any of the seven authors from Kenya? I see that all of them are based in Ghana.

Response: Yes, all authors are based in Ghana but have had previous research works conducted in Kenya. 

Please do some proofreading of the manuscript for typo and grammar errors. You wrote “marita status” in the table on page 21.

Response: Thank you for drawing our attention. We have corrected this. 

Reviewer #2: 

Thank you for the opportunity to review this manuscript. The paper is worthwhile – especially considering the persistence of IPV in sub-Saharan Africa. Below I note some comments and concerns that will help improve the paper.

Introduction

The introduction does not provide a compelling case for why the study is important, particularly within the Kenyan context. Why is Physical IPV a serious issue in Kenya? 

Response: Thank you for the comment. We have now improved the introduction section. 

What makes it endure?

Response: Thank you for the comment. We have now provided more information on this. It reads: “Despite the strides Kenya has made in establishing legal and policy frameworks to address physical IPV [11], challenges persist due to limited coordination among sectors and service providers, inadequate financial and human resources, insufficient equipment, and a lack of knowledge among service providers [12]. Additionally, flawed evidence collection practices hinder enforcement and successful prosecution efforts [12]. Consequently, most cases of physical IPV remain underreported, primarily due to financial barriers and fear of encountering an unsupportive or discriminatory response from service providers [12]. Thus, making physical IPV an enduring social canker and public health concern.”

 What are the effects of this form of IPV, etc.? 

Response: We have now provided this information. It reads: “This call is premised on evidence of a strong association between physical IPV and several adverse health effects and social risks. For instance, physical IPV has been reported to significantly increase the risk of women’s engagement in drug abuse and alcohol consumption [8]. Furthermore, physical IPV is linked to a high risk of mental health distress inkling post-traumatic stress disorders [9,10].”

Also, justify the focus on physical violence and not the other forms.

Response: We have now justified that it is the most common IPV reported elsewhere and in Kenya. It reads: “Evidence from 26 sub-Saharan African countries identifies physical violence (30.58%) as the major IPV suffered by women followed by emotional (30.22%) and sexual IPV (12.26) [5]. In the context of Kenya, approximately 35 percent of women experience physical violence orchestrated by their intimate partners [6]”. Additionally, we added the effects of physical IPV. 

The last two sentences of the first paragraph - “Globally, approximately 30 per cent of women have been exposed to intimate physical violence. While in Kenya, approximately 35 per cent of women experience physical violence orchestrated by their intimate partners”. The second sentence is incomplete. I suggest putting a comma between the two sentences might make the comparison clearer.

Response: Thank you for the comment. We have revised these sentences. It reads: “Globally, approximately 30 percent of women have been exposed to intimate physical violence [4]. Evidence from 26 sub-Saharan African countries identifies physical violence (30.58%) as the major IPV suffered by women followed by emotional (30.22%) and sexual IPV (12.26) [5]. In the context of Kenya, approximately 35 percent of women experience physical violence orchestrated by their intimate partners [6]”

Methods

Outcome variable- please specify if this is lifetime violence or just in the past twelve months.

Response: Thank you for drawing our attention. We have now indicated that it is lifetime experience of physical violence. 

Results

Please comment on some of the wide CIs e.g., age 44-491.82, 5.36

Response: Thank you for this comment. However, we, respectively, disagree that an OR of 3.13 with a corresponding CI of 1.82-5.36 is too wide as argued. The magnitude of the OR itself is a crucial factor to consider. An OR of 3.13 indicates a substantial association between the variables under investigation. While a wider CI may initially raise concerns about precision, it's essential to recognize that the point estimate still suggests a significant effect.

Discussion

The study found that 28.8% of participants had experienced physical IPV. In the introduction, the authors noted that previous studies reported an average of 35%. It is important to comment on this disparity. Could it be different data sets or time frames- What does this reduction mean?

Response: Thank you for the comment. We have now clarified this: “The prevalence of physical intimate violence observed in our study exceeds the 19.7% reported in SSA [20], yet falls below previous findings in Kenya (35%) [6] and among Zambian women (44.7%) [21]. The disparity between our estimated prevalence and that of Kimuna et al. [6] suggests potential advancements in women's assertiveness over time. Notably, Kimuna et al.'s study [6] was based on the 2014 KDHS, whereas ours reflects data from the 2022 KDHS. It is plausible that improvements in women's ability to resist physical violence from their intimate partners have occurred during this interim period. Such progress may signify evolving societal attitudes and increased awareness surrounding intimate partner violence.”

A significant finding from the study is the observed geographical variation. What explanations could account for this variation, and what implications does it carry?

Response: Thanks for your comment. A spatial regression model (GWR) has been included to show significant variables that explains the observed geographical variation.

---

## [Decision Letter · Decision Letter 1]

7 Aug 2024

Spatial distribution and determinants of Physical Intimate Partner Violence among women in Kenya: Evidence from the 2022 Kenya Demographic and Health Survey

PONE-D-24-02046R1

Dear Dr. Ayebeng,

We’re pleased to inform you that your manuscript has been judged scientifically suitable for publication and will be formally accepted for publication once it meets all outstanding technical requirements.

Kind regards,

Adobea Yaa Owusu, MA, PhD, MPH

Academic Editor

PLOS ONE

Additional Editor Comments (optional):

Reviewers' comments:

Reviewer's Responses to Questions

**Comments to the Author**

1. If the authors have adequately addressed your comments raised in a previous round of review and you feel that this manuscript is now acceptable for publication, you may indicate that here to bypass the “Comments to the Author” section, enter your conflict of interest statement in the “Confidential to Editor” section, and submit your "Accept" recommendation.

Reviewer #1: All comments have been addressed

Reviewer #2: All comments have been addressed

2. Is the manuscript technically sound, and do the data support the conclusions?

Reviewer #1: Yes

Reviewer #2: Yes

3. Has the statistical analysis been performed appropriately and rigorously? 

Reviewer #1: Yes

Reviewer #2: Yes

4. Have the authors made all data underlying the findings in their manuscript fully available?

Reviewer #1: Yes

Reviewer #2: Yes

5. Is the manuscript presented in an intelligible fashion and written in standard English?

Reviewer #1: Yes

Reviewer #2: Yes

6. Review Comments to the Author

Reviewer #1: Congratulations to the authors. My concerns about spatial variations have been addressed. The work is now publishable as authors are entitled to their opinions and argument.

However, it is a bad science to state that IPV justification increases the likelihood of IPV experience because earlier studies said so. What are you adding to science? A mere regurgitation of earlier works? The fact that previous studies have made such assumptions does not make it right. The earth was once said to be flat; new evidence emerged, and it now said to be spherical.There is new evidence that the association between the two variables is bi-directional.

The authors responded that "Thus, over time women who justify IPV may develop tolerant attitudes toward IPV violence against

women and consider the violence as normal in their life process." In other vein, I argue that, as women continue to experience IPV, they gradually justify it, just as those who use drugs will justify its use. It's a simple logic.

Since you have made up your mind, you need not rerun your analysis. However, for the sake of knowledge, test for the association between the two variables using justification of IPV as dependent variable and actual experience as independent and see what the outcome says. You do not need to include the results in the manuscript. Just check and see so that you are more informed than the earlier studies you mentioned.

Reviewer #2: The authors have addressed the comments made in the first review (or provided justifications -in some cases- for not addressing others).

My only comment which i feel needs to be addressed is that while the strength or scientific contribution of the study lies in the method (geo-spatial analysis)- the method is not sufficiently justified. The text provided lacks specificity and does not clearly show the link between geo-spatial analysis and IPV.

7. PLOS authors have the option to publish the peer review history of their article (what does this mean?). If published, this will include your full peer review and any attached files.

Reviewer #1: No

Reviewer #2: No

---

## [Editor Report · Acceptance letter]

20 Aug 2024

PONE-D-24-02046R1 

PLOS ONE

Dear Dr. Ayebeng, 

I'm pleased to inform you that your manuscript has been deemed suitable for publication in PLOS ONE. Congratulations! Your manuscript is now being handed over to our production team.

Kind regards, 

on behalf of

Professor Adobea Yaa Owusu 

Academic Editor

PLOS ONE